# Stress Effect of Food Matrices on Viability of Probiotic Cells during Model Digestion

**DOI:** 10.3390/microorganisms9081625

**Published:** 2021-07-30

**Authors:** Petra Matouskova, Julie Hoova, Petr Rysavka, Ivana Marova

**Affiliations:** 1Faculty of Chemistry, Brno University of Technology, Purkynova 118, 61200 Brno, Czech Republic; matouskova@fch.vut.cz (P.M.); xchoova@fch.vut.cz (J.H.); 2Pharmaceutical Biotechnology, Ltd., Slezska 949/32, 12000 Prague, Czech Republic; rysavka.petr@seznam.cz

**Keywords:** probiotics, food matrices, food stress, cell viability, model digestion

## Abstract

The aim of this study was to evaluate the influence of model (alcohol, sugar, salt, protein and acid) and real foods and beverages on the viability of probiotics during incubation and artificial digestion. Viability of monocultures *Lactobacillus acidophilus* CCM4833 and *Bifidobacterium breve* CCM7825T, and a commercial mixture of 9 probiotic bacterial strains, was tested by cultivation assay and flow cytometry. In model foods, the best viability was determined in the presence of 0.2 g/L glucose, 10% albumin and 10% ethanol. As the most suitable real food for probiotic survival, complex protein and carbohydrate substrates were found, such as beef broth, potato salad with pork, chicken with rice, chocolate spread, porridge and yoghurt. The best liquid was milk and meat broth, followed by Coca-Cola, beer and coffee. Viability of probiotics was higher when consumed with meals than with beverages only. Addition of prebiotics increased the viability of probiotics, especially in presence of instant and fast foods. Generally, the highest viability of probiotics during artificial digestion was observed in mixed culture in the presence of protein, sugar and fat, or their combination. The increase of cell viability observed in such foods during model digestion may further contribute to the positive effect of probiotics on human health.

## 1. Introduction

Probiotics are living microorganisms that, when administered in sufficient quantity (around 10^6^–10^7^ CFU/mL or g of carrier food product), provide a health benefit to the host, in particular through a replacement or inclusion process beneficial bacteria in the gastrointestinal tract [1,2].

In many studies, the effects of probiotics and prebiotics on various diseases have been observed. Some studies confirmed that dietary fiber and probiotics have positive effects on infectious diseases [3]. Probiotics have also shown positive responses to clinical treatment against several diseases and disorders, such as constipation and diarrhea, food allergies, inflammatory bowel disease, prevention and treatment of diabetes, obesity and cancer and diseases related to pathogenic microbes [4,5,6]. The antimicrobial activity of probiotics occurs through (i) reduced pH due to production of acetic and lactic acids, (ii) bacteriocins’ accumulation and (iii) compounds blocking bacterial adhesion to the epithelial cells and consequently reducing pathogen toxins’ production [7]. Probiotic bacteria are also crucial for the maturation of immune cells. This intestinal microbiota stimulates the maturation and functionality of the immune cells through their metabolites [5]. Probiotics are involved in the regulation of intestinal health, improved lactose digestion and maintaining bone health, and they make functional components such as antioxidants and anti-hypertensives [7]. Further, recent evidence and ongoing studies suggest that intestinal microbiota has a bidirectional effect on mood disorders and can thus affect stress and anxiety [4,8]. Research on human diseases is revealing the vital roles played by the gut microbiota. Understanding the impact of the gut microbiota on the host health is essential to design strategies focused on probiotics’ manipulation. Therefore, other knowledge on options which are responsible for increasing the viability of probiotics will allow us to design new strategies to improve the health of the consumer [9].

There are several mechanisms by which probiotics may benefit humans, including production of antimicrobial substances, strengthening of intestinal barrier, modulation of immune response and antagonism of pathogenic microorganisms, either by production of antimicrobial agents or by competition for binding sites, nutrients and growth factors [4,10,11]. More precisely, the interaction between probiotics and pathogens can be divided into three steps: the physical interaction between the probiotic and the epithelium, the interaction between probiotics and the immune system and, finally, the direct interaction between probiotics and pathogens [12]. The interaction between probiotics and pathogens may be observed in the hosts, but also in foods, where it can have a positive effect too, and incorporating probiotics in food matrices can be a new option for food safety [11,12]. Moreover, probiotic bacteria incorporated into foods should be able to survive gastric transit and reach the small intestine in sufficient numbers of viable cells [2,11].

In food-delivered probiotics, viability of probiotics is essential to achieve the health benefits associated with their consumption. Except for technological stress during food processing followed by stress connected with stabilization, packaging and storage, probiotics should undergo additional stress during consumption. The passage through the unique environment of the gastrointestinal tract is a source of high gastric acidity, oxygen stress induced by ROS released from mucosal surfaces, bile salt stress, osmotic stress and many other effects [2]. Thus, food substrates used to carry probiotics can be considered as one of the major factors in regulating colonization of probiotics in the gastrointestinal tract. Food helps to buffer the bacteria through the stomach and may contain other functional ingredients, for example prebiotics, that can influence growth, viability and survival, acid and bile tolerance, adhesion of probiotics to intestinal cells [11] and different functionality of probiotics that determine their efficacy in the gastrointestinal tract [13].

A way of increasing the efficacy of probiotic preparations may be the combination of both probiotics and prebiotics as synbiotics, which provide an improved survival during the passage of the probiotic bacteria through the intestinal tract [13]. Development of functional foods may also modulate the gut microbiota and convey health effects. Major challenges in this area can be the incorporation of probiotics in foods, selection of the prebiotic candidate or selection of bacterial strains and encapsulation of probiotic bacteria [8]. However, the addition of prebiotics to products may negatively influence the product [14]. Probiotics may also change the sensory quality of the product because some strains may grow in the food matrix and produce metabolites, which interact with the food. The encapsulation of the probiotic may also affect the food texture [15]. On the other hand, dietary supplements in the form of capsules, tablets and other formats may be used. Nevertheless, a current study on probiotics does not present a definitive answer as to whether there is superiority or equivalence on delivery of probiotics in foods or in the form of supplements [10].

However, probiotics used as supplements may reduce the functional efficacy of probiotics due to exclusion of the potential synergistic effect of the food, if they are not served together with appropriate food [13]. Many studies have focused on testing the effect of the food matrix on probiotics during storage. Regarding the incorporation of probiotics into food products, the food matrix should meet these requirements: low aw, neutral or slightly acidic pH, presence of fiber or prebiotic compounds and a high buffering capacity (for example a high fat content). For incorporation of probiotic strains into the product, the strain selection is also important [16]. However, the effect of food on probiotics during digestion should be tested as well, because the ability to survive in the gastrointestinal environment is recognized as a fundamental requisite for probiotics [17,18].

The main aim of this work was to study the influence of different types of real food and beverages on the viability and growth of probiotic bacteria during simulated passing through the gastrointestinal tract. The main goal was to mimic the most common conditions and environments used during probiotics’ consumption in a broad population, such as the effect of diverse real foods, including some homemade products, instant foods, snacks, fast food products and some beverages, to find practical recommendations for a dietary regime for optimum probiotics’ intake.

## 2. Materials and Methods

### 2.1. Material

#### 2.1.1. Probiotic Strains

Bacterial strains *Lactobacillus acidophilus* CCM 4833 and *Bifidobacterium breve* CCM 7825T used in this study were purchased from the Czech Collection of Microorganisms in Brno, Czech Republic.

The commercial preparation Biopron 9 (Walmark, Ltd., Trinec, Czech Republic) was selected as an optimum source of mixed probiotic strains. This food supplement contained a mixture of 9 bacterial cultures of *Bifidobacterium bifidum*, *Bifidobacterium breve*, *Bifidobacterium longum*, *Lactobacillus acidophilus*, *Lactobacillus casei*, *Lactobacillus plantarum*, *Lactobacillus rhamnosus*, *Lactobacillus lactis* and *Streptococcus thermophilus*, 9 × 10^9^ CFU (colony-forming units) in daily dose; additionally, the capsule contains 240 mg of fructo-oligosaccharides.

#### 2.1.2. Food Matrices

All used samples of real foods are summarized in Table 1. The set of real samples should represent typical foods and beverages consumed daily by most of the population. Beverages, instant foods (instant soups, pasta with cream sauce, porridge) and snacks (yoghurt, pudding, chocolate spread, pastry, poppy seed cake, chips, potato salad) were obtained from local retail. Instant foods were mixed with hot water or milk according to the producers’ recommendations. Homemade products (mixed vegetable broth, chicken broth, beef broth, boiled chicken meat with rice, potato salad with fried pork) were prepared immediately before experiments from natural materials (300 g) boiled in water (1 L) with 3 g of salt. Hamburger was received from fast food retail. Fruits and vegetables were used as mixtures of tomato, cucumber and pepper, with the ratio of 1:1:1.

### 2.2. Methods

#### 2.2.1. Cultivation of Probiotic Bacteria

The strains were inoculated into MRS broth (HiMedia, Mumbai, India) and incubated for 48 h at 37 °C. The cells were harvested by centrifugation at 6000 rpm for 4 °C for 10 min (Hermle Z36 HK, Hermle, Germany), and washed with distilled water. Afterwards, cells were used for inoculation, and after incubation and cultivation in different environments, for viability analysis.

Before use, the probiotic strains from the commercial Biopron capsule were hydrated by sterile distilled water for 20 min and then used for inoculation and cultivation.

#### 2.2.2. Model Food Matrices

As model food matrices, solutions with various concentrations of alcohol (ethanol 5–40%), sugar (0.2–20 g/L of glucose), salt (0.4–10 g/L of NaCl) and protein (bovine serum albumin 5–20 g/L) were prepared in sterile distilled water. As a model of acidic food, sterile distilled water adjusted to pH = 3 by HCl was used. Standard and pure chemicals were purchased from Sigma Aldrich (Sigma-Aldrich, Merck KGaA, Darmstadt, Germany). All used model foods are summarized in Table 2.

### 2.3. Model Digestion

Artificial stomach juice was prepared from 0.25 g of pepsin dissolved in 100 mL of distilled water. To this solution, 0.84 mL of 35% hydrochloric acid was added. Final pH was adjusted to 0.9. Artificial pancreatic fluid was prepared with 0.25 g of pancreatin and 1.5 g of sodium hydrogen carbonate in 100 mL of distilled water (pH = 8.9). Bile fluid was composed of 0.4 g of bile acid salts dissolved in 100 mL of phosphate buffer. Probiotics were added to 300 g of beverage, liquid food or solid food homogenized in an appropriate water volume. A mixture of probiotic cells (1.10^10^ CFU) and food/beverage was first incubated at 37 °C for 15 min to reach starting values of cell numbers. Then, the mixture of probiotics and food matrix was mixed with stomach juice (in the ratio of 3:1) and incubated for 20 min at 37 °C. Additionally, intestinal fluids formed by a mixture of pancreatic fluid and bile salts in the ratio of 1:1 (*v*/*v*) were added to a final ratio of food and all digestion juices of 3:2, similarly to GIT conditions. Model digestion was set as a shortened continuous process.

### 2.4. Determination of Cell Viability by Cultivation Method and Flow Cytometry

The viability of probiotic bacteria was tested by cultivation assay on Petri dishes with MRS medium by the overflow method. After 48 h of cultivation, colony-forming units, e.g., bacterial colonies (CFU/mL), were counted. The cultivation method was used for evaluation of all types of foods and beverages, including model foods.

The viability of probiotic bacteria exposed to model and real liquid foods and beverages was also followed by direct analysis of cell viability using flow cytometry (Apogee Flow Systems, Hemel Hempstead, UK). As a fluorescent probe, propidium iodide (Sigma-Aldrich, Darmstadt, Germany) was used. For analysis, 100 µL of liquid sample was taken, diluted to 1 mL and added to propidium iodide. After 5 min of incubation in darkness, viability of probiotic cells was measured in a clear liquid model and real food matrixes in each stage of digestion.

### 2.5. Viability of Monocultures and Mixed Probiotic Cultures in Model and Real Foods and Beverages in Conditions of Model Digestion

Probiotic bacterial cultures of *Lactobacillus acidophilus*, *Bifidobacterium breve*, or a commercial mixture of probiotic cells (1 capsule of Biopron 9; about 9.10^9^ CFU) were added to the 300 g of model or real food/beverages in the amount of 1.10^10^ CFU. Each of the tested probiotics samples were first mixed with food/beverage and incubated for 15 min at 37 °C. Then, the samples were exposed to model digestion according to Section 2.3. In regular intervals (after or before incubation), the amount of living cells was measured using cultivation methods and flow cytometry. The evaluation of viability, CFU number and, thus, the influence of individual model and real foods is complicated by the continuous growth of probiotic cells in any environment. Thus, the number of viable cells was measured gradually at the beginning of each stage of continuously performed model digestion and after its completion. The first evaluation was performed after the incubation of probiotics with food matrix as the response to dilution and food stress environment. This measurement was followed by evaluation after acidification at the beginning of stomach digestion, then after 20 min incubation in stomach juice (pH 2.1), followed by cell counting after alkalization and mixing with the mixture of pancreatic and bile juices and at the end of the digestion process (pH 7.9, 2 h). Acidity of the environment (pH) was checked during the whole digestion process.

### 2.6. Viability of Mixture of Probiotic Cells in Different Types of Foods and Beverages with/without Addition of Prebiotics

Probiotics from a commercial capsule were re-hydrated for 20 min and solubilized in sterile distilled water. Then, cells were mixed with real food matrix and with or without the addition of prebiotics (300 mg of inulin, Sigma-Aldrich, Merck KGaA, Darmstadt, Germany). The recommended daily dose of probiotics mixture (1 capsule, 1.10^10^ CFU) was suspended in 300 g of homogenized food or beverage. This prepared mixture was then incubated at 37 °C for 15 min. Then, samples were taken for analysis of the number of viable cells at the beginning of the digestion process (1 mL). Due to the heterogenic character of samples, viability of probiotic cells was determined using cultivation techniques. Determining the number of cells in the samples was performed after 48 h of cultivation in MRS agar.

### 2.7. Viability of Monocultures and Mixed Probiotic Cultures in Real Foods and Beverages in Conditions of Model Digestion

Probiotic bacterial cultures of *Lactobacillus acidophilus*, *Bifidobacterium breve*, or a commercial mixture of probiotic cells (content of capsule Biopron 9) were added to the 300 g of model or real food/beverages in the amount of 1.10^10^ CFU. At the beginning of incubation, model food matrixes can be considered as sterile (see Section 2.2.2). Real foods were prepared immediately before incubation and, thus, the presence of other microorganisms was minimized. Each of the tested probiotics samples were first mixed with food/beverage and incubated for 15 min. Then, the samples were exposed to model digestion according to Section 2.3. In regular intervals (after or during incubation), the amount of living cells was measured. The viability of probiotics during simulated gastrointestinal conditions were performed using cultivation methods and flow cytometry (liquid samples). Cultivation methods were used for all heterogenic samples, and for homogenous solutions, it was possible to determine them directly by FC. The evaluation of viability and CFU in individual model and real foods are complicated by continuous growth of probiotic cells in any environment. Thus, the viability after digestion (or individual stages) was compared to the value of CFU at the beginning of model digestion, after short-time incubation with the food.

## 3. Results

### 3.1. Viability of Probiotics in Model Food Matrices

The aim of the present study was to evaluate the viability of tested probiotic cultures in the presence of selected model and real foods and beverages in model conditions of the digestive tract. As a model food, solutions with various concentrations of alcohol, sugar, salt, protein and acetic acid were prepared. Real foods were selected according to the most common dietary preferences in the population.

In all model foods except acidic food, the pH value was set to pH 7, and the model acid food is of pH 3. In Table 3, pH values in tested real beverages and foods are presented. With the exception of juice (pH 3) and beer (pH 5.1), all tested beverages and foods exhibited pH in the range of 6.0–7.3. Thus, we cannot expect some dramatic influence of pH of tested food matrixes on viability of probiotic cells. However, a more acidic pH (approximately 5.5–6.5) will probably be more suitable for probiotic bacteria growth.

#### 3.1.1. Growth of Mixed Probiotic Culture in Model Foods Exposed to Artificial Digestion

The influence of basic model foods on concentration and viability of probiotic cells during artificial digestion was studied (Figure 1A–F). As the probiotic culture, a mixed commercial preparative was used. The measurement was performed using flow cytometry, and as model foods water (water-based food), acetic acid solution of pH 3 (acid food), glucose (sweet food), protein (protein-based food), sodium chloride (salted food) and ethanol (alcohol-based food) at different concentrations (see Table 2) were used. The evaluation of viability and CFU in individual model and real foods is complicated by the continuous growth of probiotic cells in any environment, thus, in some samples, an increase of CFU after model digestion was observed. It is necessary to note that the number of CFU determined by flow cytometry is related to 1 μL of sample because of the extreme sensitivity of this method when compared with cultivation techniques.

First, model digestion using the probiotic mixture without addition of food matrix was measured. Probiotic capsule content was re-hydrated in water and placed directly into the stomach and then to intestine fluids. This experiment simulated a model swallowing an intake of dry probiotic capsule without the addition of any food or beverage. A significant decrease of the number of living cells was observed after model digestion without the presence of some food matrix. After drastic pH changes, a dramatic decrease (approximately 17.5×) of the number of originally present cells was recorded at the beginning of the intestinal stage of model digestion (Figure 1F). On the other hand, after model digestion of probiotics in the presence of water, an approximately 12 times increase of the original number of cells was recorded. It is a model of swallowing the intake of a probiotic capsule together with 300 mL of water. Subsequently, the effect of protein (Figure 1B) on cell growth during digestion was tested. Two hours after the addition of intestinal juices, a significant increase of cell concentration in sample with the addition of 10 g/L of bovine albumin was observed. In this sample, about 3 times more than the original number of cells was determined.

The effect of salt environment on cell growth during digestion is introduced in Figure 1C. After a moderate increase of CFU in the environment of intestinal fluids, two hours after addition, a slight decrease of cell concentration was observed (except the lowest concentration used). Moreover, with increasing salt concentration, the more intensive decrease of cell concentration was observed. For 0.4 g/L NaCl concentration, the final CFU value after two hours of model digestion was about 2 times higher than the original CFU value found at the beginning of the exposition to artificial digestion.

Next, the effect of alcohol (Figure 1E) on cell growth during digestion was monitored. After the addition of intestinal juices, a moderate increase of CFU was detected, but after a longer digestion, a decrease of cell concentration was observed again. The decrease in cell number with increased alcohol concentration was observed as well. Nevertheless, a low ethanol concentration (5–10%) exhibited a relatively good effect on CFU (about 3–4x increase after complete model digestion).

Further, the effect of model food with glucose on cell growth during digestion was monitored (Figure 1A). The highest increase of cell number was recorded in the sample with 2 g/L of glucose. After two hours of model digestion, about 5 times higher CFU in this sample was determined. Finally, the number of cells in acidic model food was tested (Figure 1E). At the beginning of the intestinal stage, a decrease of the number of cells in acidic food was observed. However, after two hours of exposition to the intestinal juice, the CFU number for acidic food was already 2 times higher.

Generally, according to total CFU numbers in model foods, the most suitable environments for probiotic preparative intake were recognized as: (i) water, (ii) protein at the concentration of 10 g/L, (iii) glucose at 2 g/L and low ethanol concentration (5–10%). The most negative effect was exhibited predominantly by salt, which was probably caused by osmotic stress, similar to the effect of higher glucose concentrations. A negative effect of ethanol as a bacteriostatic agent was observed after long-term exposition to higher concentrations (20% and more), while lower concentrations of ethanol exhibited positive effects on probiotic viability. The reason could be a cross-response to the external stress effect of ethanol and probably also an ability of some probiotics to use ethanol as an additional carbon source.

#### 3.1.2. Flow Cytometry Determination of Viability of Probiotics Exposed to Model Digestion in Liquid Environments

In the following section, the influence of model foods on the CFU number and viability of probiotic monocultures of *Lactobacillus acidophilus* and *Bifidobacterium breve* after model digestion was studied (Figure 2). For comparison, a mixed probiotic culture was used. The scheme of the experiment is described in Section 2.5. After artificial digestion in stomach juice followed by incubation in intestinal juices, the CFU number was directly determined by flow cytometry. At the end of incubation, the culture of *Lactobacillus acidophilus* exhibited a high increase of CFU predominantly in model food containing protein, but also slightly in other model foods (Figure 2A). Conversely, the strain *Bifidobacterium breve* grew intensively, predominantly in model foods containing saccharide (Figure 2B). In the commercial probiotic mixture, the highest increase of CFU number was found after the model digestion in the presence of water at pH = 7 and in the presence of acidic solution (pH = 3). Furthermore, the CFU increase was detected in foods containing protein and 10% alcohol as well (Figure 2C).

Flow cytometry was also used for determination of cell viability in the environment of clear (centrifugation 5000 rpm, 10 min) meat and vegetable broth prepared according to Section 2.1.2.

Results are illustrated in Figure 3. When compared with model foods, monocultures as well as mixed probiotics culture exhibited similar trends, corresponding to some of the typical model foods, such as protein (meat broth) or sugar (vegetable broth). The best environment for all tested samples was chicken broth, containing high concentrations of proteins. Except for in this environment, monocultures grew quite slowly (approximately 4–5x lower biomass formation) when compared with the mixed culture. Growth of mixed culture in meat extract was lower than in the presence of 10 g/L of protein.

The resistance of the probiotic mixture against acidic pH mentioned above is probably caused by the addition of some protective agents to the preparative, which could prevent damage of cells by the acidic stomach environment, as described by the producer. A similar study described a neutral or slightly acidic pH (5–6) as a suitable food environment for protection of lactic acid bacteria [19]. This pH was optimal for growth of probiotics during food manufacturing and cell survival during storage and digestion. Water activity can also improve cell protection during digestion, as well as solids and gels. The presence of highly fermentable sugars and fiber also promoted survival of cells during food storage and digestion. High fat content exhibited positive effects on cells, mainly due to good buffering capacity [19].

Flow cytometry is a highly sensitive method of cell viability determination, especially when compared to the cultivation techniques (Appendix A). About a 1000 times lower amount of sample was sufficient for direct evaluation of cell viability (see Appendix A). We can suppose that clear beverages (water, tea, meat broth) and model solutions are suitable for flow cytometry measurements of cell viability, in contrast to colloid materials (e.g., milk), food homogenates and all environments containing particles. Thus, the comparison of all tested real beverages was performed by cultivation techniques to eliminate the interference of impurities.

### 3.2. Influence of Real Foods and Beverages on Viability of Probiotics

#### 3.2.1. Growth and Viability of Probiotics in Real Beverages Exposed to Model Digestion

In the next part of this work, the viability and growth of probiotic bacteria *Lactobacillus acidophilus, Bifidobacterium breve* and a commercial mixture of probiotic strains was tested after passing through the digestive tract, with a wide range of different types of food and beverages. Tested products can be divided into the following groups: drinks, soups, main courses and snacks (Table 1).

Traditionally, the use of probiotics in dairy beverages has been widely extended. However, since people who suffer from allergy to milk proteins or have severe lactose intolerance cannot consume dairy beverages, non-dairy beverages such as fruits, vegetables and cereals juices may also represent a suitable vehicle to deliver probiotics to consumers, with regard to the stability of the cells during storage [18]. In this study, water, black tea, coffee, beer, juice, Coca-Cola and milk were tested. In all these beverages, the highest amounts of surviving cells and increased growth of probiotic cells were observed in the intestinal environment (Figure 4). These results were also verified by flow cytometry. It should be mentioned that in contrast to the other tested beverages, only the environment of beer showed an increase of the mixed probiotic culture viability during the digestion step in the stomach. In all other beverages, a significant decrease of viability after 20 min of incubation in stomach juice was measured (Figure 4, 2nd and 3rd column).

Flow cytometry was also used for the study of the influence of real beverages on the CFU number and viability of probiotic monocultures of *Lactobacillus acidophilus* and *Bifidobacterium breve* after model digestion, in comparison with the probiotic mixture studied (Figure 5). After artificial digestion in stomach juice followed by incubation in intestinal juices, the CFU number was determined in monocultures and the probiotic mixture.

When the monocultures of *Lactobacillus acidophilus* or *Bifidobacterium breve* were used, a significant increase (about 2.5–3×) of surviving cells was observed only in the milk environment (Figure 5A). This finding confirms the fact that milk is a well-known suitable environment for probiotic bacteria. The probiotic cells from commercial preparation (Figure 5B) showed the highest increase of living cells in a Coca-Cola and milk environment. The result whereby increased CFU numbers of probiotic cells after passing through artificial digestion have also been observed in the presence of coffee, beer and black tea was also interesting. The mixture of probiotic strains is probably more stable in different environments when compared with monoculture.

#### 3.2.2. Growth and Viability of Probiotics in Some Real Liquid and Solid Meals during Exposition to Model Digestion

Real meals were processed according to Section 2.1.2 and divided into two groups: (i) liquid meals (e.g., soups—instant and homemade) and (ii) solid meals and snacks (see Table 1). The influence of all these environments on the viability of monocultures of *L. acidophilus* and *B. breve* as well as on the viability of probiotics in mixed culture was studied (Figure 6).

Regarding homogenates of tested soups (broth, pea soup, beef soup) (Figure 6), the number of viable probiotic cells of monocultures as well as the commercial mixed culture after model digestion was similar in the presence of all tested soups. The group of tested solid foods was formed by hamburger, instant pasta with cream sauce, fried pork meat with potato salad and chicken with rice as main courses (Figure 6). Here, the highest number of viable probiotic cells of monocultures as well as commercial mixed culture after model digestion was found in the presence of potato salad with pork, porridge and chicken and rice. These meals are mainly rich in proteins, sugars and milk (in the case of porridge) (Figure 6).

In the group of snacks (Figure 6), the absolutely highest CFU number of mixed probiotic culture after model digestion was observed in the presence of chocolate spread. Increased CFU number in probiotic mixture was also observed in the presence of mixed fruits, chocolate pudding with whipped cream, salted chips, fruits and yoghurt (Figure 6). As in the previous group, these foods exhibited similar characteristics, mainly regarding high sugar content. It could also be important that the presence of fat, milk and, surprisingly, also salt stress in the presence of salted chips was accepted by probiotics relatively well. Overall, similarly to beverages, the mixed probiotic culture was more stable in all tested food environments compared with monocultures (Figure 6).

#### 3.2.3. Viability of Mixed Probiotic Culture in the Presence of Prebiotics and Real Foods and Beverages

Finally, the addition of prebiotics and its influence of viability of the mixture of probiotics in the presence of different types of foods/beverages was observed (Figure 7). As a prebiotic in this study, inulin was used as one of the most studied and widely used prebiotics [10]. The mixed probiotic culture from the capsule was incubated for 20 min in the presence of individual foods with and without the addition of inulin.

As predicted, the highest number of survived probiotic cells was recorded in the presence of milk. A higher growth of probiotic cells compared to growth in distilled water was determined for fruits, yoghurt, fruit juice, black tea, Coca-Cola and beef broth environments. When the probiotic mixture together with a prebiotic was used, a similar effect was observed, and the highest cell growth predominantly in the presence of milk, beef broth and Coca-Cola was determined (Figure 7). The addition of the prebiotics/probiotics mixture led to an increase of viability of probiotics in the presence of some “unhealthy” processed foods, such as hamburger and Coca-Cola.

## 4. Discussion

Nowadays, probiotics belong to the most popular food supplements and ingredients. They can be taken in different ways and forms, and can be a part of different dietary regimes. A minimum dose of 10^6^ colony forming units per mL or g (CFU/mL or CFU/g) must be reached for the food product to be labelled as probiotic [18]. Some researchers even suggest increasing the dose up to 10^7^ CFU/mL or CFU/g [19], because the viability of microorganisms is the key to achieve the health benefits. Viability depends on the environmental condition in the final product and its interaction with the probiotic strain. Thus, the food matrix’s chemical composition and its physical state can substantially affect growth, stability and survival of probiotic microorganisms during digestion [19]. Microbial cultures should be capable of growing in substrate media, survive during technological processing and maintain their viability throughout storage.

For stability and viability of probiotics, not only the type of probiotic preparative and processing, but also the foods and beverages ingested simultaneously with probiotics can substantially influence the final effect of probiotics. Moreover, differences in the concentration and viability of probiotic cells when passing through the gastrointestinal tract may be observed. The present work focused mainly on the influence of different types of model and real foods/beverages on the viability of probiotic bacteria during digestion. The aim was to select foods that can increase cell viability during artificial digestion, which may further contribute to the positive effect of probiotic cells on human health.

Species of *Lactobacillus* and *Bifidobacterium* are the most commercial probiotics available in the food market [18]. Therefore, in this work, the influence of food and beverages was tested on monocultures of *Lactobacillus acidophilus*, *Bifidobacterium breve* and for comparison, also on a commercial mixture of probiotics containing 9 different probiotic strains.

To reach probiotic status and ability to promote health benefits, it is necessary to evaluate cells’ resistance to the digestion process [10]. In this study, probiotic cultures were incubated for a short time in selected foods and beverages, and then the viability of the probiotics was tested in a model artificial digestive tract. The conception was proposed to find suitable and, conversely, completely inappropriate food taken into the body together with probiotics. First, incubation of probiotics in the environment of various model foods was performed, followed by artificial digestion. As a model food, solutions with various concentrations of alcohol, sugar, salt, proteins and acid were prepared. Viability of probiotics in model solutions was measured directly by flow cytometry, which is a highly sensitive method sufficient for clear beverages and model solutions, in contrast to colloid materials, food homogenates and all environments containing some particles, which should be evaluated by cultivation techniques.

Furthermore, real foods and beverages were tested in a similar way as model foods. Recently, it was found that some food matrices are more protective than others during storage and simulated gastrointestinal conditions [4,10,18]. In particular, attention has been paid to dairy products such as cheese, yogurt and fermented milk. Cheese has a potentially good matrix for delivery of probiotics due to several characteristics, including its higher pH value, greater buffering capacity, greater fat content and nutrient availability and lower oxygen content. Additionally, fruits and vegetables have been found to be an ideal addition to probiotic foods, likely because they provide essential nutrients for bacterial growth [4], and fiber-rich products such as fruits and grains can also increase the viability of probiotic bacteria during storage and simulated gastrointestinal conditions [18,19].

The tolerance of probiotic bacteria to gastric and small intestine conditions seems to be significantly influenced by the food carrier. Many studies showed that vegetable matrices could improve probiotics’ vitality during the gastric transit. The good protection of probiotics against simulated gastric juices was confirmed for carrot juice, similarly to dairy matrices [2]. In other studies, the food matrix impact on lactic acid bacteria viability during food digestion of dairy versus non-dairy products was compared [18]. Comparing milk and fruit juices, the high tolerance for bile acids and pepsin in the milk environment was observed. Regarding different flavors of juice, banana and carrot juice exhibited highly positive effects when compared with orange juice [4]. Very good positive effects on probiotics similar to dairy products were reported for pasta [7]. When comparing different dairy products, the best results were obtained with cheese, milk and yoghurt, and finally, ice cream [13]. However, for different strains, differences in the order of these matrices were recorded [19]. In the case of milk as one of the most suitable environments for probiotics, it should be mentioned that milk belongs to the group of highly allergenic foods, causing both lactose intolerance and allergy to milk proteins. In milk, lactose is the main, but not the only, C source for probiotics. Lactose content is substantially lowered in fermented milk products and the rest can be utilized by probiotic bacteria, which further contributes to minimization of lactose’s effect in the intestinal tract. Regarding milk protein allergies, milk products should be fully removed from nutrition of sensitive subjects and replaced by some plant sources, which are suitable as probiotics carriers, as documented in this study [20,21].

Interesting effects on probiotics’ viability were found in another type of real beverage—beer. In contrast to the other selected beverages, only the beer sample showed an increase of the mixed probiotic viability during the digestion step in the stomach (see Figure 4). The reason could be that beer has some unique properties different than other tested beverages. In this study, typical Czech beer Pilsner Urquel was used. This beer is characterized by high antioxidant activity, ethanol content of 4.9% and a high content of residual extract containing a mix of various compounds with potential prebiotic effects. Further, Czech beer exhibited a high buffering capacity in the digestive tract, which can positively influence the viability of probiotics. In our previous work, the effects of Czech beer on human health, pH changes in an artificial stomach and intestinal juice after the addition of beer (ratio 1:3; *v*/*v*) were measured. In stomach juice mixed with Pilsner Urquel beer, pH increased from 1.45 to 2.95 and proteolytic activity was about 1.4× higher. In contrast, pH of intestinal juice dropped from an initial pH of 8.80 to 7.25, and protease activity was about 2.5× higher after beer addition [22]. Such specific effects of beer on stomach and duodenum environments could positively influence the viability and growth of probiotics during model digestion.

The main purpose of this work was to test the potential stress effect of food matrices on the growth of probiotics and the possible protection against the effects of the digestive system. This study contains a set of original data represented by results of incubation of probiotics in the presence of many different types of complex food matrixes and exposed to model digestion. Meals containing a combination of plant and animal foods such as meat, pasta, cream and some of the most popular beverages were tested. Some of our data agree with previously published studies focused on the influence of fruit and dairy products [4,7,18,19,23]. Based on the results, it can be concluded that the best way appears to be the combination of mixtures of probiotics with foods rich in proteins and sugars, and the best option among beverages seems to be milk, except for people with intolerances and allergies. Nevertheless, other beverages such as tea, coffee, Coca-Cola and beer can be recommended for consumption with probiotics. Further, we can conclude that probiotic microorganisms have survived better in the presence of meals, when compared with consumption with the beverages only. Higher growth of probiotic cells was observed in foods containing high concentrations of sugar, protein and fat, or their optimal combination. These data are also confirmed by a recent study, showing that carbohydrates are the most widely used protective compounds during dehydration, storage and exposure of probiotics to the gastrointestinal tract [16]. The addition of prebiotics can positively influence the effect of processed and fast foods (hamburger, Coca-Cola) on probiotics’ viability in the presence of these foods.

## 5. Conclusions

The present work dealt with the influence of different types of model and real foods on cell viability during digestion. The main goal of the work was to analyze the stress effect of different food environments on the viability of probiotic monocultures (*Lactobacillus acidophilus* and *Bifidobacterium breve*) and a commercial probiotic mixture during model digestion. The highest growth of *L. acidophilus* cells was detected in foods containing protein, while *B**. breve* exhibited the best growth in the presence of saccharides. The best growth of the mixed probiotic culture was determined in tap water at neutral pH and in acidic environments, as well as in the presence of foods containing protein and 10% alcohol.

Regarding real foods and beverages, the best food environments for increasing probiotic viability were complex meals with meat and fiber, milk products—porridge and yoghurt, chocolate spread and mixed fruits. Among beverages, the best option was milk, but Coca-Cola, coffee, beer and black tea were also acceptable for the viability of probiotics. We can conclude that probiotics are more viable when consumed with meals compared to the beverages only. The viability of the mixed probiotic culture was higher in all environments when compared with both monocultures. The addition of prebiotics further positively increased the viability of probiotics, even in the presence of processed foods. In general, the highest viability of probiotics during artificial digestion was observed in the mixed probiotics culture in the presence of protein, sugar and fat, or their combination. The increase of cell viability observed in such foods during model digestion may further contribute to the positive effects of probiotics on human health.

## Figures and Tables

**Figure 1 microorganisms-09-01625-f001:**
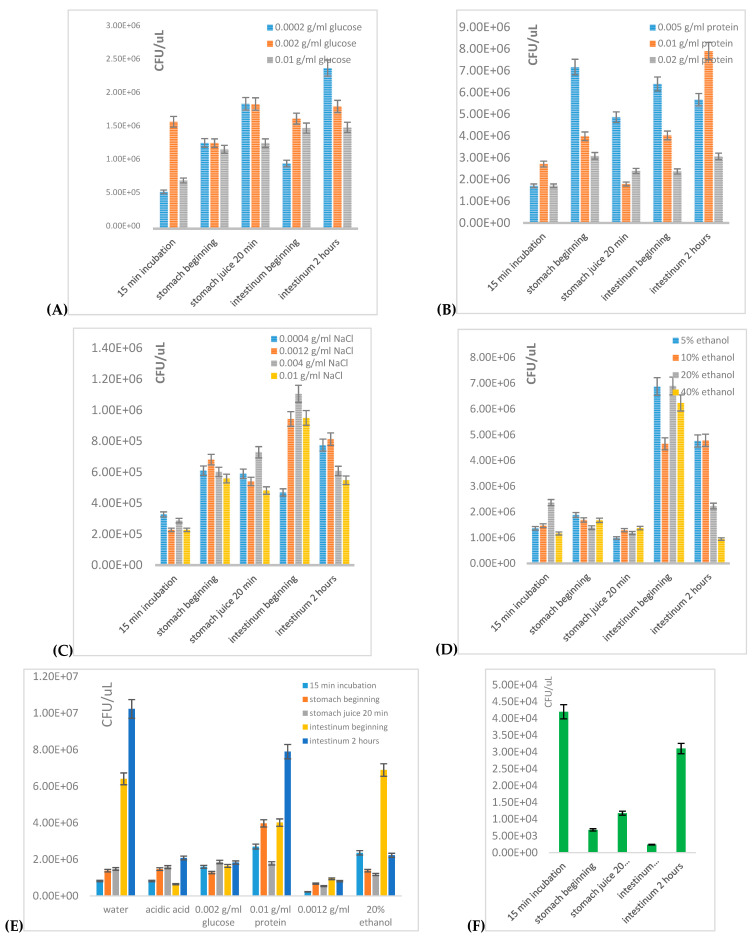
Viability of mixed culture of lactic acid bacteria during the process of digestion of model foods containing probiotics (commercial probiotic mixture): (**A**) sugar concentration, (**B**) protein concentration, (**C**) salt concentration, (**D**) alcohol concentration, (**E**) comparison of model factors, pH 3 was set by acetic acid, and (**F**) incubation of cell suspension directly in digestive fluid.

**Figure 2 microorganisms-09-01625-f002:**
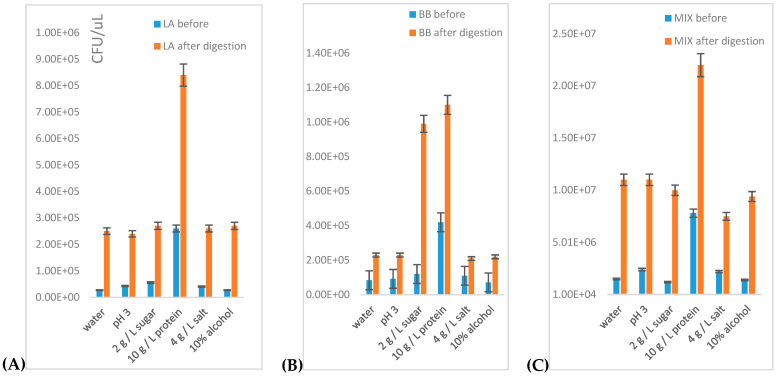
The influence of model foods and the type of probiotic on the number of viable cells after model digestion: (**A**) *Lactobacillus acidophilus*, (**B**) *Bifidobacterium breve* and (**C**) probiotic mixture.

**Figure 3 microorganisms-09-01625-f003:**
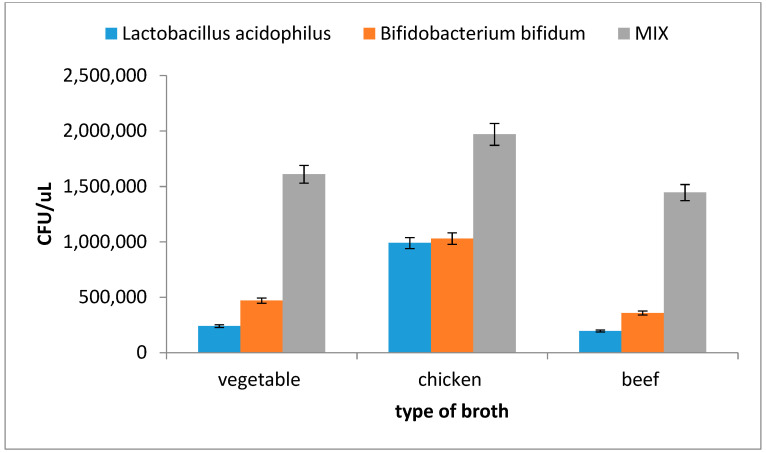
The influence of the type of food extract (broth) and the type of probiotic on the number of viable cells after model digestion.

**Figure 4 microorganisms-09-01625-f004:**
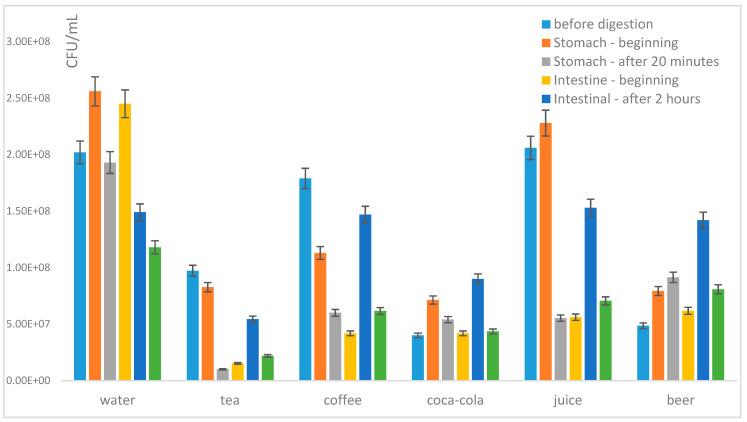
Viability of mixed probiotic culture in selected beverages during the gradual process of model digestion.

**Figure 5 microorganisms-09-01625-f005:**
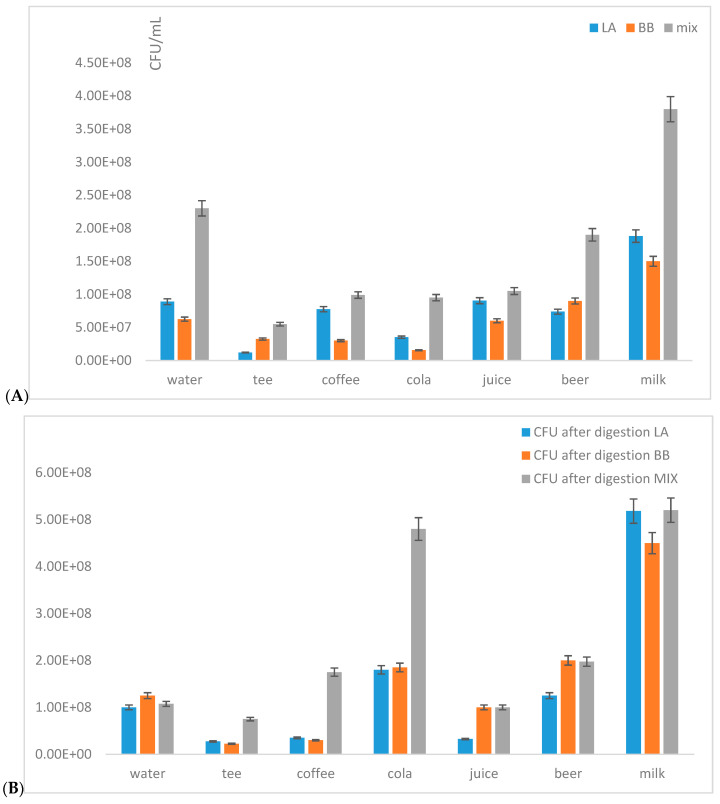
The influence of different types beverages and the type of probiotic on the number of viable cells after model digestion: comparison of monocultures *Lactobacillus acidophilus* (LA) and *Bifidobacterium breve* (BB) and mixed probiotic culture (MIX). (**A**) Viability after 15 min of incubation, and (**B**) viability after complete model digestion (2 h).

**Figure 6 microorganisms-09-01625-f006:**
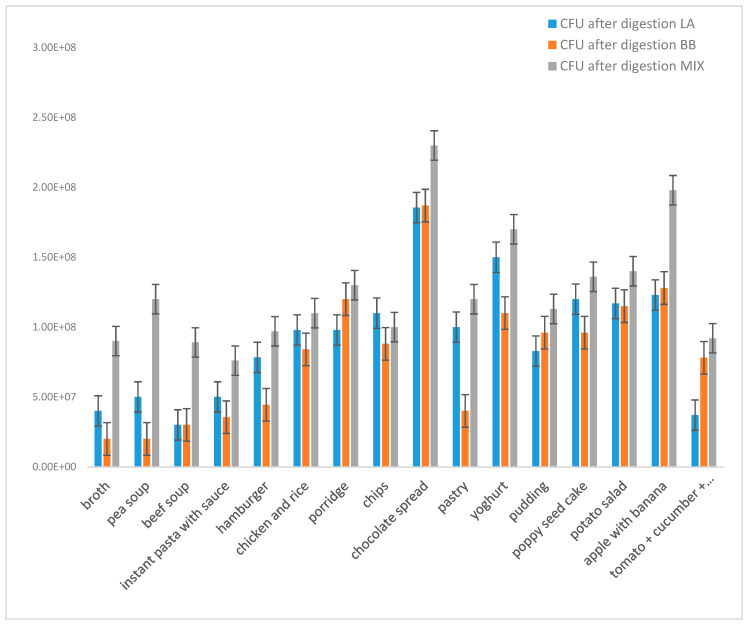
The influence of different types of food and the type of probiotic on the number of viable cells after model digestion (CFU/mL): comparison of monocultures *Lactobacillus acidophilus* (LA) and *Bifidobacterium breve* (BB) and mixed probiotic culture (MIX).

**Figure 7 microorganisms-09-01625-f007:**
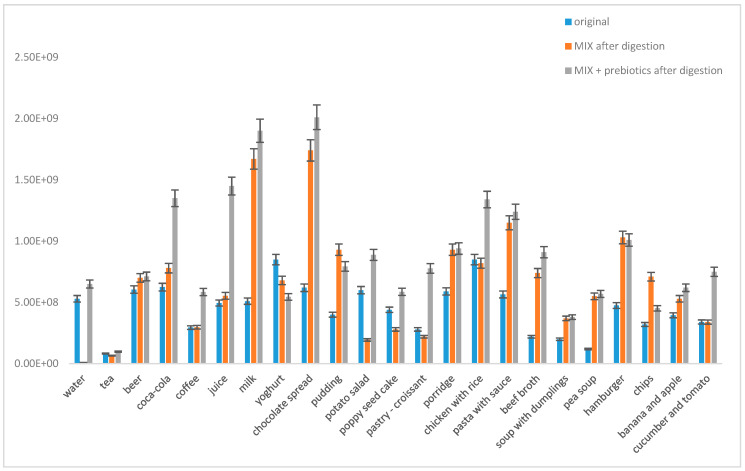
The effect of various foods and the addition of prebiotics (inulin) on the viability of probiotic cells in the mixed probiotic preparative (MIX).

**Table 1 microorganisms-09-01625-t001:** Overview of tested real foods and beverages.

Beverages	Soups	Main Courses	Snacks	Fruit, Vegetables
tap water	beef broth instant	Hamburger—fast food	yoghurt chocolate	
black tea	pea instant soup with croutons	pasta with cream sauce—instant	pudding with whipped cream	apple with banana
black coffee	instant soup with liver dumplings	chicken with rice—boiled	chocolate spread	tomato, cucumber and white pepper
beer (4.9% alcohol)	vegetable broth—homemade	potato salad with fried pork	porridge	
juice orange	chicken broth—homemade		pastry	
milk (3.5% fat)	beef broth—homemade		poppy seed cake	
Coca-Cola			Chips—salted	

**Table 2 microorganisms-09-01625-t002:** Overview of tested model foods.

	1	2	3	4
Sterile distilled water	pH 7	pH 3 (with HCl)		
Protein (Bovine Albumin)	5 g/L	10 g/L	20 g/L	
Saccharide (Glucose)	0.2 g/L	2 g/L	10 g/L	
Salt (NaCl)	0.4 g/L	1.2 g/L	4 g/L	10 g/L
Alcohol (Ethanol)	5%	10%	20%	40%

**Table 3 microorganisms-09-01625-t003:** pH values in real beverages and foods.

Beverage/Liquid	pH	Food	pH
Water	7.0	Pasta with cream sauce—instant	7.1
Black tea	6.9	Chicken with rice—boiled	7.2
Black coffee	6.0	Potato salad with fried pork	6.7
Beer (4.9% alcohol)	5.1	Hamburger—fast food	7.1
Juice, orange	3.2	Yoghurt—chocolate	6.2
Milk (3.5% fat)	7.1	Pudding with whipped cream	7.3
Coca-Cola	5.6	Chocolate spread	6.3
Beef broth	6.2	Porridge	5.2
Vegetable broth	6.0	Pastry	7.0
Chicken broth	6.3	Poppy seed cake	6.2
Beef broth instant	6.2	Chips—salted	7.1
Pea instant soup with croutons	6.4	Fruits (apple + banana)	6.7
Instant soup with liver dumplings	6.5	Vegetable mix	5.8

## Data Availability

The study did not report any data.

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
