# Peer review of "Stress Effect of Food Matrices on Viability of Probiotic Cells during Model Digestion"

_microorganisms, 2021, doi:10.3390/microorganisms9081625_

Round 1

Reviewer 1 Report

The paper is well written and follows a precise and linear line. Address the issue in a careful and detailed way. The experiments are well conducted and with good results. Some typos need to be corrected, and some sentences refined. It would add in the discussion some insights on the properties of milk and its derivatives, adding references to pathologies related to lactose and the role of probiotics. Uniform the style and size of the tables.

Author Response

Dear Colleagues,

In attached file I am sending response to reviewers of revised version of the manuscript entitled:  " Stress effect of food matrices on viability of probiotic cells during model digestion” by the authors P. Matouskova, J. Hoova, P. Rysavka and I. Marova (as corresponding author).

This work was prepared to publication in Special Issue of the Microorganisms Journal -  "Microbial Stress Response as a Tool for Biotechnology”.  

Reviewer 2 Report

This paper describes the impact of foods on mono- and multistrain probiotic viability by using representative ingredient models and real foods as materials, which have been evaluated under simulated digestive conditions as main method.  It is an interesting topic, and the manuscript is well introduced with a state-of-the art particularly well written. The section "materials and methods" is right described, including tables that overview the numerous tested ingredient model and real food systems used in the experiments. The authors report and discuss some interesting results before concluding: which food matrix appear better than others with regards to the viability of mono- and multistrain probiotics under digestive model conditions. 

Even if the manuscript is quite good, some additional information is needed to specify in order to improve the discussion of the results: 

• It is important in this study to provide the ratios of each probiotic in the mixed commercial preparation, used as comparison sample, and the strains should be specified, if possible. Each strain is unique and specific, so its proportion and contribution in the mixture may significantly affect the global viability results that have been evaluated with only one cultivation method;
• Cell viability is one of the most important parameter of probiotic benefits for health. However, the contribution of the dead cells compared to living ones at each stage of digestion may be significant, and deserves to be considered and discussed (postbiotic effects) as indicator;
• Please cite a reference reporting the negative effect of prebiotic addition on the product (p.2 in the introduction, line 7, §3);
• Please specify the type of sterile water used (e.g. distilled or tap water), provide its resistivity as indication;
• The authors said: "water activity can also improve cells protection during digestion", please specify in which direction, lower or higher aw?
• In Fig.4, only beer sample shows the increase of the mixed probiotic viability during the digestion step in stomach. This should be announced in the result and discussed. How do the authors explain such results?

Moreover, the discussion and conclusion sections should be improved. In the discussion, the phrases "In this study, probiotic cultures were...by cultivation techniques" (p. 14-15) are not necessary, these repeat the description of the methods used. It would be more valuable to discuss some results such as the beer effect, in opposite of the other selected beverages, on the mixed probiotics during the digestion simulation in the stomach.
In the conclusion section, the two first paragraphs should be rewritten and condensed in one phrase as a general state as the nature of food ingredient model and real systems impact the mono- and multistrain probiotic viability. 

Finally, some errors are found in the manuscript:

1. Introduction:
- suppress "in" (§1, line 2)
- was => were (§2, line1)
- suppress "in" (last §, line 2)

2. Materials and Methods
- In Table 2. distiller => distilled
- jodide => iodide (last §, line 5, p.4)

3. Results
- ty => by (§2, line 2)
- Figure 5. A & B. tee => tea and cola => coca-cola (x-axis legend)
- probiotics => prebiotics (p. 12, subtitle 3.2.3)
- cocs-cola => coca-cola (p.12, last §)

4. Discussion
- Include "ingredients" between food and supplements (line 1, §1)

Author Response

(The authors gave the same response as above.)
